# Focal Muscle Vibration (fMV) for Post-Stroke Motor Recovery: Multisite Neuroplasticity Induction, Timing of Intervention, Clinical Approaches, and Prospects from a Narrative Review

Alessandro Viganò [1], Claudia Celletti [2], Giada Giuliani [3], Tommaso B. Jannini [4], Francesco Marenco [1], Ilaria Maestrini [4], Rosaceleste Zumpano [5], Edoardo Vicenzini [3], Marta Altieri [3], Filippo Camerota [2], Vittorio Di Piero [3,†] and Massimiliano Toscano [3,6,*,†]

1. IRCCS Fondazione Don Carlo Gnocchi, 20148 Milan, Italy
2. Physical Medicine and Rehabilitation Division, Umberto I Hospital, 00185 Rome, Italy
3. Department of Human Neurosciences, "Sapienza" University of Rome, 00185 Rome, Italy
4. Department of Systems Medicine, University of Rome Tor Vergata, 00133 Rome, Italy
5. Department of Chemistry and Drug Technologies, "Sapienza" University of Rome, 00185 Rome, Italy
6. Department of Neurology, Fatebenefratelli Hospital—Gemelli Isola, Isola Tiberina, 00186 Rome, Italy
* Correspondence: massimiliano.toscano@uniroma1.it; Tel.: +39-06-49914682; Fax: +39-06-49694254
† These authors contributed equally to this work and share the last authorship.

**Abstract:** Despite newly available therapies for acute stroke and innovative prevention strategies, stroke remains the third leading cause of disability-adjusted life-years (DALYs) lost worldwide, mostly because more than half of stroke survivors aged 65 and over exhibit an incomplete functional recovery of the paretic limb. Given that a repeated sensory input is one of the most effective modulators of cortical motor and somatosensory structures, focal muscle vibration (fMV) is gaining growing interest as a safe, well-tolerated, and non-invasive brain stimulation technique to promote motor recovery after stroke with a long-lasting and clinically relevant improvement in strength, step symmetry, gait, and kinematics parameters. In this narrative review, we first summarize the structural (neural plasticity) and functional changes (network relearning) triggered by the stroke lesion and carried out at a brain and spinal cord level in an attempt to recover from the loss of function. Then, we will focus on the fMV's plasticity-based mechanisms reporting evidence of a possible concurrently acting multisite plasticity induced by fMV. Finally, to understand what the most effective fMV rehabilitation protocol could be, we will report the most recent evidence regarding the different clinical approaches and timing of the fMV treatment, the related open issues, and prospects.

**Keywords:** stroke; focal muscle vibration (fMV); repetitive muscle vibration (rMV); non-invasive brain stimulation (NIBS); neuroplasticity; adaptive plasticity; motor recovery; post-stroke spasticity; cortical hyperactivity; motoneuron excitability

## 1. Motor Recovery after Stroke and Neuroplasticity: From Stroke Lesion to Network Relearning

In the last three decades, stroke has become a dramatic fast-growing burden in the world. From 1990, the absolute number of cases increased substantially as well as incident strokes (70.0%), deaths from stroke (43.0%), prevalent strokes (102.0%), and disability as expressed by disability-adjusted life-years lost—DALYs (143.0% DALYs) [1]. Despite newly available therapies for acute stroke and innovative prevention strategies, stroke still remains the second-leading cause of death, with an estimated global cost of over US$ 721 billion due to incomplete recovery [2] and frequent comorbidities [3–5].

Post-stroke motor impairment represents a great part of this burden since, to date, stroke remains the third leading cause of disability, and more than half of stroke survivors aged 65 and over exhibit an incomplete functional recovery after stroke with reduced mobility [2].

Motor recovery after stroke consists of improvement in two domains [6,7]. The first is the so-called "true recovery", which refers to the improvement of body function and structures, and the second is the compensation, which indicates the patient's ability to accomplish a goal through adaptation of remaining elements or substitution with a new approach. While the latter also depends on implicit and explicit compensatory strategies, true recovery mainly depends on neuroplasticity [8,9], which consists of the brain's natural property to reorganize, changing properties, structures, and pathways to acquire or improve skills.

Neuroplasticity can be defined as "adaptive plasticity" if the subtended plastic changes facilitate full recovery of an involved function, whereas, in the case of incomplete recovery or the occurrence of unwanted symptoms (e.g., pain, compensatory movement patterns, and delayed-onset involuntary movements) [10], it is called "maladaptive plasticity" [11–15]. Specifically, a growing body of evidence from neuroimaging and neurophysiological studies shows that incomplete motor recovery after stroke may be the result of maladaptive structural and functional changes in perilesional and remote regions triggered by the focal brain lesion. These maladaptive changes may lead to functional disconnection in the apparently intact perilesional brain areas and to an altered balance of excitatory and inhibitory influences within the motor network, both in the affected and unaffected hemisphere [10,16–18]. It is worth noting that even though maladaptive changes hinder functional recovery, they also represent the brain's attempt to restore a function loss through neuroplasticity and thus are susceptible to recovery through plasticity-based rewiring [18].

In this scenario, understanding both the structural and functional changes that occur soon after a stroke and the mechanisms underlying motor recovery, represents the fundamental basis for optimizing rehabilitative interventions that can condition the dysfunctional cerebral structures and network by enhancing the adaptive plasticity and even modulating the maladaptive one [10,19,20].

After a brain injury such as a stroke, the plasticity-based attempt to recover the function loss involves three distinct but partially overlapping phases [21]: reversal of diaschisis with cell genesis and repair, functional neuronal plasticity (e.g., changing properties of central monoaminergic neuronal pathways) [22,23], and neuroanatomical plasticity (i.e., the capability to establish and consolidate new neural networks in response to a change in the environment) [24]. Even though these changes occur both at a cortical and a peripheral level, they mostly involve the cortex, the ideal site for the plasticity to take place [25].

To describe in detail all the mechanisms underlying brain plasticity goes beyond our purpose. Therefore, we will mainly focus on synchronous electrical hyperactivity that represents a measurable change observed in several cortical areas as an immediate response to stroke and, most importantly, it can be susceptible to modulation by non-invasive brain stimulation techniques (NIBS) such as focal muscle vibration (fMV).

Being responsible for the facilitation of activity-dependent plastic change, cortical hyperactivity represents the electrical expression of neuronal plasticity, which depends on the peculiar ability of neocortical neurons to modify their response properties following prolonged alteration in input activity [26].

Early cortical hyperactivation can promote post-stroke plastic changes by enhancing the synaptic activity that produces a long-lasting increase in signal transmission between two neurons (i.e., long-term potentiation, LTP). The GABA receptor's downregulation and the NMDA receptor's binding site enhancement, both linked to LTP-related cellular plasticity together with metabotropic and AMPA glutamatergic receptors, have indeed been described as peculiar to the hyper-activated damaged cortex.

Besides synaptic remodeling, synchronous electrical hyperactivity can also promote axonal sprouting, which in turn plays a pivotal role in promoting neuroanatomical plasticity, i.e., the capability to recruit additional cortical regions to establish and consolidate new neural networks and to even facilitate subsequent refocusing towards a shifted sensorimotor cortical representation [24,27].

In the specific, the ischemic lesions induce both long-distance cortico–spinal axonal sprouting [28] and horizontal axonal sprouting between previously non-connected areas [29].

From a functional and even "teleological" point of view, the role of changes in perilesional and remote brain regions (i.e., in both the affected and in the unaffected hemisphere) triggered in the very acute phase by the focal brain lesion, as well as the role of the recruitment of remote or secondary brain structures, is not completely understood [16,17,30].

In general, this compensatory recruitment is not "maladaptive" because stroke patients with greater motor impairment show stronger recruitment of secondary brain structures, and the disruption of these areas through transcranial magnetic stimulation (TMS) has demonstrated that their recruitment is functionally significant [31]. Nevertheless, it leads to an incomplete recovery [32] because stroke patients with poorer recovery have a stronger (i.e., more widespread and often bilateral) cortical over-activation and a greater amount of previously "silent" region recruitment [33]. A possible explanation is that the projections from ipsilateral neurons located in non-primary motor areas are less numerous and less efficient at exciting spinal cord motor neurons than those from M1 [27,34,35]. Moreover, the integrity of the lesioned hemisphere's motor cortex (ipsilesional M1) is related to better post-stroke motor recovery [34,36,37], so the modulation of ipsilesional M1 became one of the most effective targets for rehabilitation therapy [38,39]. In this sense, repetitive focal muscle vibration (fMV) represents a very effective neuro-rehabilitative intervention that can induce prolonged changes in the excitatory/inhibitory state of the primary sensorimotor cortex directly acting on the paretic limb.

## 2. Focal MUSCLE Vibration (fMV) for Post-Stroke Motor Recovery: Concurrent Multisite Plasticity-Driven Mechanisms and Clinical Correlates

The first description of the therapeutic use of muscle vibrations dates back to 1892, when the great neurologist Jean-Martin Charcot delivered the milestone lecture titled *"Vibration therapeutics: Application of rapid and continuous vibrations to the treatment of certain nervous system disorders"*. However, the motor effects of fMV in patients with various types of central motor disorders became a research topic only in the late twentieth century, thanks to the pioneering studies by Hagbarth and Bishop (for a historical overview see Celletti et al.) [40].

In the last two decades, the attempt to improve motor impairment through the modulation of plasticity-based mechanisms that are responsible for motor recovery after stroke has become a pivotal research field.

In this scenario fMV (also known as repetitive muscle vibration—rMV, local or segmental vibration) [41–47] is achieving growing interest as a safe and well-tolerated NIBS technique able to condition the stroke-related dysfunctional cerebral structures and pathways and promote long-lasting motor recovery (see Appendix A).

The rationale behind the fMV treatment is that muscle vibration represents a strong proprioceptive stimulus and, when repetitively delivered at a high frequency and low amplitude (see Section 3), can induce plastic changes that are responsible for cortical motor and somatosensory structure modulation, changes in synaptic responsiveness, and late sustained reorganization of the synaptic pathways.

Both the clinical improvement and the related neurophysiological changes induced by the fMV treatment outlast the mere period of stimulation itself. Reportedly, they are still present after 30 min and several months after the stimulation, respectively, depending on the adopted stimulation protocol (e.g., vibration frequency and amplitude, duration and number of sessions), the chosen evaluation parameter (e.g., cortical area map/volume, H-reflex and MEP amplitude), and the outcome evaluated (e.g., improvement in strength, step symmetry, spasticity, discriminative ability, motor planning, execution, etc.) [48–51].

This represents the strongest evidence that inducing activity-dependent neuroplasticity constitutes the main mechanism underlying the clinically relevant recovery of the paretic limb reported in stroke patients after the fMV treatment. Namely, fMV can promote both

central and peripheral plasticity by modulating cortical excitability (central action) and motoneuron excitability (peripheral action), respectively.

Stroke lesion entails concurrent central and peripheral changes that in turn are responsible for the impairment of different "central and peripheral aspects" of motor function, such as strength, step symmetry, spasticity, discriminative ability, motor planning, execution, etc. Since fMV can improve several of those motor aspects through plasticity-based rewiring, it is expected that the fMV can concurrently promote plasticity in the SNC and the spinal cord by simultaneously modulating cortical excitability and motoneuron excitability.

In this view, a recent $^{99m}$Tc-HMPAO SPECT and neurophysiological case study reported concurrent fMV-dependent changes in cortical perfusion and motoneuron excitability that were related to a clinically relevant improvement in patient's strength, step symmetry (with reduced limping), and spasticity after fMV treatment [52].

For greater expository clarity, the plasticity-driven central and peripheral mechanisms of fMV will be treated separately in the next paragraphs along with the respective clinical correlates.

### 2.1. fMV-Induced Plasticity in the Brain

A repetitive focal muscle vibration, when applied during a voluntary isometric contraction, can induce prolonged changes in the excitatory/inhibitory state of the primary motor cortex in healthy subjects [46] and chronic stroke patients [45], thereby leading to long-term changes in motor performance of the affected limb through intrinsic plasticity-related mechanisms.

Given the specific pattern of direct connections linking SI and M1, the repeated sensory input delivered by the fMV can produce Ia fiber afferent input that reaches M1 [53–56]. This strong proprioceptive input can change the M1 excitatory/inhibitory state through long-term synaptic potentiation (LTP), as demonstrated by increased motor map areas of the vibrated muscle. fMV also reduces the resting motor threshold and induces changes in short-interval intracortical inhibition (SICI) and sensorimotor interaction patterns [45,46,49,50,57,58].

The strong and powerful hyperactivation of muscle proprioceptors induced by fMV can also produce LTP-like plastic changes in perilesional brain regions and modulate their connections to the spinal cord. Thus, the LTP-like plastic changes induced by fMV do not involve the sole M1 but probably entail entire network relearning achieved through the modulation of the effective connectivity (i.e., the direct or indirect influences that one brain region exerts over another one) [59], both in the affected and the unaffected hemisphere [16,17].

Regarding the restitution of ipsilesional effective connectivity in the damaged hemisphere, the clinically relevant improvement described in stroke patients after fMV treatment is related to a reduction in the post-stroke abnormalities of the corticospinal excitability [45] and changes in perfusion in the affected hemisphere [52], which is as a marker of reactivation of functionally impaired circuits within the motor network [60]. Taken together, these data suggest that fMV can act on the modulation of the intracortical inhibitory systems to enhance the intra-hemispheric cortico–cortical connections between differently specialized cortical areas after stroke.

Concerning the contralesional effective connectivity, it should be noted that fMV increases the effectiveness of cortical–spinal connections. Given that the secondary motor areas (e.g., the dorsal premotor area—PMd) have prominent bilateral connections to the spinal cord [32], the activity-dependent modulation of the cortical–spinal connection could in turn modulate the contralesional hemisphere's hyperexcitability (i.e., interhemispheric imbalance), whose role in motor recovery after stroke may be crucial [35,39]. In this view, it is intriguing that in healthy subjects [61] and stroke patients treated with fMV, the clinical improvement was also related to changes in excitability and perfusion in the unaffected hemisphere [52,62]. To date, it is unclear whether fMV can also modulate the pathological

transcallosal influences originating from the unaffected hemisphere, thus acting on the contralateral effective connectivity.

### 2.2. fMV-Induced Plasticity in the Spinal Cord

Operant conditioning of ipsilesional sensorimotor cortex excitability is not the sole plasticity-based mechanism involved in the fMV-related motor recovery after stroke.

fMV can induce activity-dependent changes in the intrinsic spinal cord properties at the Ia-motoneuron synapse level and in specific spinal cord circuits depending on the muscle vibrated. In this view, a concurrent induction of plasticity in the spinal cord could explain why fMV, differently from the other plasticity-based rehabilitation techniques, is very effective even on subcortical and spinal damage [63] and in reducing post-stroke spasticity as well.

In the subacute and chronic stages of stroke, the velocity-dependent increase in muscle tone is mainly due to the stretch-sensitive (spastic) muscle overactivity [64], implying that muscle contracture causes excessive responsiveness to stretch, which in turn aggravates contracture and impedes voluntary motor neuron recruitment. Postsynaptic inhibitory circuits [65] and recurrent inhibition [66] are dampened in patients with spasticity, supporting the concept that decreased postsynaptic inhibition is involved in the hyperexcitability of the stretch reflex along with depressed presynaptic inhibition [67].

Delivering a repetitive high-frequency (usually 100 Hz, see Section 3) focal vibration at this stage of the disease can lead to a reduction in post-stroke spasticity since it stimulates the muscle spindles and evokes Ia afferent discharge, inducing synaptic events, changes in synaptic responsiveness, and late sustained reorganization of the synaptic pathways.

The activity-dependent decrease in the intrinsic motoneuron excitability represents the strongest evidence for the plasticity to be directly induced in the spinal cord by fMV treatment [45,49,50,57,68]. Namely, evidence from studies on healthy subjects [49,50,69,70] suggests that fMV can promote spinal plasticity mainly through presynaptic changes in the spinal cord. From a neurophysiological point of view, the fMV-induced presynaptic results in a decrease in the amplitude of the H reflex (i.e., the electric analog of the spinal stretch reflex), which is the most adopted paradigm for investigating the plasticity induced in the human spinal cord [68].

In subacute and chronic stroke patients treated with fMV, operant conditioning of H-reflex drives the inhibition of pathological motor patterns [33,57,71–73] and relates to a long-lasting and clinically relevant improvement in the functional ability of the paretic limb among several motor aspects, including strength, spasticity, gait, and kinematic parameters [40,41,44,45,57,72–77], as well as motor planning, execution, and perception [50].

Beyond brain and spinal cord changes, vibration can alter muscle homeostasis affecting directly several endocrine–metabolic parameters. Most of this evidence derives from whole-body vibration paradigms [78,79], but it could be inferred that to some extent it could also occur after focal muscle vibration.

At a supraspinal level, the reduction in post-stroke spasticity after fMV treatment may be also related to changes in the third phase of reciprocal inhibition (RI-3), which is responsible for the descending control of spinal interneurons and correlates with the post-stroke hypertonicity degree [80]. Although the fMV seems to leave RI-3 unchanged in healthy subjects [49], RI-3 is affected by cortical modulation so that, in chronic stroke patients, fMV might indirectly modulate RI-3 by acting on the altered supraspinal inhibitory circuits [81].

Finally, it should be noted that in the paretic limb, the neural insult itself causes a reduction in voluntary motor unit recruitment since the acute phase, so spasticity can also develop in the early time course of stroke [82]. Mainly because the increase in muscle tone is rare and of a mild entity in the acute stage of the disease, there is poor evidence so far of a clinically relevant improvement of spasticity in acute stroke patients treated with fMV [47].

*2.3. Timing of Intervention: New Evidence for an Extended Critical Time Window*

The magnitude of the brain's natural capacity to reorganize, changing properties, structures, and pathways after a change in the environment such as a stroke (i.e., adaptative neuroplasticity), changes over time and progressively declines as time passes from the acute event. Since the adaptative plasticity drives the "true recovery" after stroke (see Section 1), the progressive decline of neuroplastic capabilities explains why improvements in motor function after stroke are time-sensitive as well, being maximum in the acute and subacute phases (i.e., within 15 days and 3–6 months after stroke, respectively), almost completed within 10 weeks post-stroke with a plateau one to six months after stroke onset depending on the motor impairment degree [83,84].

In this view, it is a crucial neurorehabilitation challenge to identify the so-called "*critical time window*" for recovery, which represents a period of heightened physiological neuroplasticity in which environmental/behavioral experience results in major changes in structure and connectivity within the brain, so that patients are more sensitive to treatments able to induce plasticity, such as fMV.

Recent studies aimed at investigating the duration of the critical window of enhanced neuroplasticity to determine the optimal period after stroke for intensive motor training found an optimal time window from 60 to 90 days after stroke and up to 6 months (i.e., the subacute phase of stroke), with smaller but significant effects in the acute phase (i.e., within 30 days) and no effect in the early and late chronic stages of the disease (i.e., 6 months after stroke onset or later) [85–87].

Few studies have been performed so far treating patients in the subacute phase of stroke with fMV and are mostly focused on gait and kinematic analysis [88–90]. Two distinct studies carried out in sub-acute stroke patients reported fMV-related improvements in motor function and spasticity, as assessed with the modified Ashworth scale (MAS) [73,91].

A large part of the studies aimed at investigating fMV effects on post-stroke motor recovery include patients treated in the late-chronic phase of the disease (≥1 year from stroke). Since they widely reported significant fMV-related improvement of several motor aspects, including strength, spasticity, gait, and kinematic parameters [40,41,44,45,57,72–77], they indirectly show that neuroplasticity is still present 1 year after stroke.

In this view, Ballester et al. [8] investigated the plasticity-based "true recovery" in hemiparetic stroke patients treated with a specific rehabilitation training program in different stages of the disease. They showed an improvement in body function and structure even at the late chronic stage, with a gradient of enhanced sensitivity to treatment that extended beyond 12 months post-stroke.

Regarding "the other side of the window", treating stroke patients in the acute phase through modulating neuroplasticity presents pro and contra and may be challenging from several points of view.

On one hand, the role of changes in perilesional and remote brain regions triggered by the focal brain lesion in the very acute phase is not completely clear [16,17,30]. Moreover, data from both VECTORS [92] and AVERT [93] trials showed that intensive training too soon may result in lesion expansion and worse motor recovery.

On the other hand, the structural and functional changes within the motor network that drive the "true recovery" occur in the immediate few hours after stroke, and most of the post-stroke rehabilitation guidelines suggest starting the rehabilitation program in the acute phase, as soon as possible. Thus, it seems to be crucial to try to further extend the critical time window to the acute phase and to understand how to modulate the first and most powerful plasticity-based brain attempt to restore function loss.

In the recent CPASS trial, even though the optimal period for heightened plasticity was 60~90 days after stroke, data showed smaller but significant improvement in the upper limb recovery of stroke patients treated in the acute phase (i.e., ≤30 days from stroke), so that the authors concluded that there might be a point in the acute phase when stroke patients are equally sensitive to treatments able to induce plasticity, such as fMV. In this regard, only one RCT has been performed so far treating stroke patients with fMV in the

acute phase [47]. It showed that fMV intervention carried out at a mean time from stroke of $43.9 \pm 18.9$ h can improve motor outcomes in a cohort of stroke patients regardless of the different baseline clinical status or the different stroke characteristics.

## 3. fMV's Different Current Clinical Approaches

fMV's capability of inducing synaptic plasticity using the proprioceptive pathway, as well as its effectiveness in neurorehabilitation [94], deeply depends on vibration parameters and treatment protocols. A recent review by Kolbasy et al. [58] shows that operant conditioning of sensorimotor cortical hyperactivity through fMV drives opposite results based on vibration frequency and amplitude. There is also great variability in the literature regarding the duration, number, and frequency of fMV sessions, the muscle(s) to treat, and even the status of the vibrated muscle [40,41].

Given that Ia afferents can fire synchronously with vibration frequencies up to 80–120 Hz [95,96], 100 Hz is the most adopted frequency of vibration used in the literature [40,41,43,45–47,49,50,57,75].

Even though some authors used both higher frequencies (i.e., 120 to 300 Hz) [44,77,89,90] and lower frequencies (i.e., 70 to 91 Hz) [73,76,89,90,97,98], it is widely recognized that delivering a 100 Hz vibratory stimulus can induce plastic changes in the brain and the spinal cord through the operant condition of cortical hyperactivity and motoneuron excitability, respectively [40,41,43,45–47,49,50,57,75]. Moreover, a vibration frequency of 100 Hz is appropriate for stimulating the muscle spindles and evoking the "spindle driving" phenomenon, which entails that Ia afferent discharge is driven at the same stimulation frequency so that it is probably the most effective frequency to induce change in synaptic responsiveness and drive late sustained reorganization of the synaptic pathways [50,99–101].

The most adopted vibration amplitude (i.e., the peak-to-peak sinusoidal displacement generated by the vibration) is from 0.2 to 0.5 mm [40,41,43,45–47,57,97], but also, in this case, there is great variability in the literature, with reported stimulation amplitudes of 10 μm [77,89,90], 15 μm [88], 1 mm [72,73], and even higher [44,74,75].

Most authors seem to agree that the optimum amplitude of the vibratory stimulus is less than 0.5 mm because such a low amplitude can activate Ia afferents by entraining the discharge rate of primary muscle spindle endings. Otherwise, a greater value tends to lead to an overflow of the stimulus into the surrounding tissues and elicit the tonic vibration reflex [40,49,102], which represents a tonic reflex contraction due to the amplification of Ia synaptic inputs induced by persistent inward currents (PICs) (see Section 4).

fMV protocols differ across the studies also depending on the pathology and the therapeutic goal. In stroke patients, treated muscles for the upper limb may include a single target muscle [75,76]; two target muscles such as the flexor carpi radialis (FCR) and the biceps brachii (BB) [45,47,90] or the pectoralis minor and the biceps brachii [43]; and up to three muscles treated simultaneously.

To treat post-stroke spasticity, authors have differently chosen to treat the extensor or flexor muscles, but whether it is more effective to apply fMV over the spastic muscles or on its antagonistic remains controversial so far [103,104].

Regarding upper limb treatment, several studies applied fMV over the wrist flexor muscles alone [45,57,73,75,76,90] or in combination with synergic muscles, such as the biceps brachialis [45,90], over the biceps brachialis coupled with the pectoralis minor [43,57], or over the hand [72]. To improve flexor muscle spasticity, other studies applied fMV over extensor muscles [44,88,89], such as the triceps brachii alone [75] or in association with the extensor carpi radialis [44].

To improve the lower limb's strength and spasticity, as well as stability, walking, and kinematic parameters, fMV was applied over the peroneus longus and the tibialis anterior [89], the Achilles and the tibialis anterior tendon [88], the rectus femoris [47], or over the quadriceps femoris, the triceps surae, and the hamstring muscles [52].

Concerning the simultaneous contraction of the vibrated muscle, recent evidence suggests that, due to homeostatic metaplasticity mechanisms, the fMV after-effects also

depend on muscle contraction during fMV [49,50]. Most of the studies don't use muscle recruitment during local vibration applications and usually, during the treatment, the subject is only asked to maintain a voluntary steady contraction of the target muscle at 20% of the maximal voluntary contraction (MVC), as assessed by visual EMG feedback [45,46,52], because voluntary muscle activity increases response to vibration probably through fusimotor co-activation and a subsequent increase in spindle discharge [49,105].

Also, there is no agreement in the literature about the number and the duration of fMV sessions required for motor recovery to be maximally effective. Excluding vibration applied synchronously to evaluation, one of the most adopted application protocols consists of delivering fMV for three consecutive days, with each daily application consisting of three consecutive 10 min sessions for each muscle, interspersed with a 1 min break [43,45–47,49,57]. Other adopted protocols include delivering fMV through a daily 60 min session for three days [45], a 50 min session three times a week for 4 weeks followed by another 12 sessions over 4 weeks [89], 30 sessions (30 min each) over 6 weeks [88], or 10 sessions (60 min each) over 2 weeks [90].

In this regard, no comparison study has been performed to address which is the most effective duration of treatment. However, since the amount of the induced change grows gradually as conditioning trials continue over subsequent days and weeks [68,106], probably the longer the stimulation period, the higher the motor recovery. Moreover, prolonged stimulation can allow the activation and transduction of proteins, and the repetition over consecutive days allows optimal memory consolidation and long persistence of the training-independent sensory learning [50].

## 4. Open Issues and Future Prospects

Focal muscle vibration (fMV) can promote post-stroke motor recovery by inducing multisite concurrent neuroplasticity to condition the stroke-related dysfunctional structures and pathways. Even though most of the studies have addressed fMV effects in combination with conventional rehabilitation programs, findings in chronic stroke patients were also detected in the absence of any other physical therapy [50]. Patients are usually required to perform at least a minimal isometric voluntary contraction but, in general, fMV has no reported contraindication or side effects and represents a safe and well-tolerated non-invasive brain stimulation (NIBS) technique, easy to perform at the bedside and able to promote post-stroke motor recovery in every stage of the disease.

To date, there are several open issues and further RCTs are needed to understand which is the most effective fMV rehabilitation protocol to reach a clinically relevant improvement of the paretic limb in different motor aspects, such as strength, step symmetry, balance and gait control, spasticity, discriminative ability, motor planning, execution, and perception.

Following the milestone studies on fMV [45,46,57], one of the most adopted protocols consists in delivering high-frequency (100 Hz) and low amplitude (0.2–0.5 mm) repetitive focal vibration stimulus for at least 3 consecutive days. These parameters are probably the most effective for operant conditioning of both motoneuron excitability and H-reflex, as well as modulating the descending control of spinal interneurons.

It should be noted that many authors adopted this stimulation protocol also to avoid the tonic vibration reflex (TVR), which represents a tonic reflex contraction due to the amplification of Ia synaptic inputs induced by persistent inward currents (PICs). Namely, since the spindle primary endings are stimulated harmonically up to 80 Hz, a 100 Hz stimulation (at a low fixed amplitude) avoids the amplification of Ia synaptic inputs induced by persistent inward currents and does not generate the TVR.

In this regard, it is not clear so far whether TVR activation is needed to induce clinical effects. On the one hand, the produced involuntary contraction in the vibrated muscles is the mechanism exploited to treat the antagonist of the spastic muscle to decrease the spasticity of the paretic limb [73]. In this regard, whether it is more effective to apply fMV over the spastic muscles or on its antagonistic muscles remains controversial so far [103,104]. On the other hand, the involuntary contraction of the vibrated muscle may lower the reflex

threshold and inhibit the antagonist muscle, so avoiding TVR may help normalize reflex excitability, decrease co-contraction, and improve motor control [94].

Even though there are no studies aimed at addressing this specific issue, most of the studies adopted a stimulation protocol (i.e., 100 Hz frequency and 0.2–0.5 amplitude) that can avoid the TVR.

Few studies have investigated differences in the fMV-related motor improvement depending on stroke characteristics, such as stroke type (i.e., ischemic or hemorrhagic stroke), localization (cortical or subcortical), extension, and side of the lesion, which could influence the efficacy of the fMV treatment [40]. Thus, is still unclear, to date, if there is a stroke subpopulation that can especially benefit from fMV treatment. Given that patients with a cortical stroke lesion have a lower motor threshold, larger map sizes and volumes, and stronger disinhibition, Marconi et al. [45] reported that the fMV-induced effect could vary depending on whether the stroke is cortical or subcortical. A stroke involving the subcortical white matter can also result in cortical deafferentation causing dysfunction within widespread cortical areas outside of the infarction. Thus, differences in stroke characteristics may influence the amount of motor recovery and cortical reorganization and make the efficacy of any given treatment vary by the specific pattern of brain damage in individual patients [107]. This is particularly true for NIBS techniques, like fMV, that depend on plasticity-based cortical reorganization for motor recovery being achieved.

Despite these methodological issues, promoting post-stroke recovery through fMV opens exciting scenarios, mostly concerning the possibility to extend the critical time window of treatment or to increase its effects.

Regarding the extension critical time window, it is worth noting that heightened physiological neuroplasticity seems to be still present 1 year after stroke [8], allowing the fMV-induced improvement of body function and structure even at the late chronic stage. However, the adaptative plastic changes within the motor network that drive the "true recovery" occur soon after stroke, and the acute phase is probably the period of maximally heightened plasticity after stroke.

Thus, in our opinion, one of the main goals of future RCTs is to individuate a "critical therapeutical point" in the acute phase where it is possible to act early on subtended plastic changes that facilitate full recovery (i.e., adaptive plasticity) without strengthening the very early maladaptive cellular and functional changes that hinder the functional recovery even though they are made in an attempt to restore a function loss (i.e., maladaptive plasticity).

Finally, thanks to the facilitation of synaptic plasticity along the proprioceptive pathway and in the central motor area, fMV induces complex and unexpected motor effects that outlast the stimulation period and also involves untreated muscle [50]. The duration and the magnitude of the elicited effects constitute a common limit of all the non-invasive brain stimulation techniques that act on brain plasticity, so their increase represents a pivotal goal to achieve.

In this regard, the advance in the knowledge of the mechanism underlying cortical- and synaptic-induced plasticity may allow designing new strategies to promote motor recovery through fMV. Similarly to the glutamatergic synapse plasticity, the adaptive plasticity induced by fMV shows dependency on the timing of the pre- and post-synaptic events (i.e., spike timing-dependent plasticity (STDP) principle [48]). The STDP principle implies that the strength of an input to a neuron depends on whether the input spike occurs immediately before or soon after the output and represents the basis of the so-called "metaplasticity", a fundamental rule of synaptic plasticity that describes how the threshold for activity-dependent synaptic plasticity is dynamic [49].

Thanks to homeostatic metaplasticity mechanisms, fMV can promote sudden and complex after-effects only when the vibrated muscle is in a state of mild, voluntary isometric contraction. Muscle contraction during fMV increases post-synaptic activity in spinal motor neurons and represents a preconditioning that in turn, through this synaptic learning mechanisms, can induce plastic changes that are responsible for the fMV-related after-effect being long-lasting and higher than expected [49,50].

Moreover, metaplasticity implies that synaptic plasticity can be modulated and even reversed by prior synaptic activity and thus represents the rationale for NIBS techniques to be used as priming stimulation to facilitate the effect of a subsequent treatment [48,107–109].

Perassi et al. [51] consolidated the fMV cortical aftereffect by combining the vibration treatment with a concomitant high-frequency 5-Hz repetitive transcranial magnetic stimulation (rTMS) protocol. They found a persistent pattern (i.e., up to 30 min) of unbalanced M1 excitability between vibrated muscle and its antagonist.

Even though very few studies have been carried out on this issue so far, applying a NIBS (e.g., rTMS or tDCS) as a priming stimulation to precondition the fMV treatment could open interesting scenarios.

Further RCTs are required to understand how to enhance the already clinically relevant effect on post-stroke motor recovery exerted by fMV, a technique known to induce multisite concurrent neuroplasticity and to promote clinical and neurophysiological changes that already last for at least 2 weeks after the end of treatment.

**Author Contributions:** A.V.: conceptualization, original draft preparation; C.C.: manuscript writing and literature review (clinical approaches); G.G.: manuscript writing and literature review (neuroplasticity); T.B.J.: manuscript writing and literature review (neuroplasticity); F.M.: manuscript writing (post-stroke spasticity); I.M.: manuscript review (fMV and spinal cord); R.Z. rebuttal and revised version preparation; E.V.: manuscript review (fMV and brain); M.A.: manuscript review (post-stroke motor impairment); F.C.: supervision, manuscript review, literature review; V.D.P.: supervision, manuscript writing and editing, manuscript review, literature review; M.T.: conceptualization, original draft preparation, manuscript review, literature review. All authors have read and agreed to the published version of the manuscript.

**Funding:** This research received no external funding.

**Data Availability Statement:** Data sharing not applicable.

**Conflicts of Interest:** The authors declare no conflict of interest.

## Appendix A

Articles were reviewed without time limits on several databases (PubMed, EMBASE, Scopus, Web of Science, and the Cochrane Library) until March 2023 for fMV. The search strategy included the following terms: (local OR focal OR repetitive) AND (vibration OR muscle vibration) AND (Stroke OR ischemic OR hemorrhagic). Other relevant articles were selected using manual identification within references of screened publications. Unpublished articles were not considered in this study.

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
