# Peer review of "Focal Muscle Vibration (fMV) for Post-Stroke Motor Recovery: Multisite Neuroplasticity Induction, Timing of Intervention, Clinical Approaches, and Prospects from a Narrative Review"

_vibration, doi:10.3390/vibration6030040_

Round 1

Reviewer 1 Report

The paper is well presented and seems quite interesting. The topic of focal muscle vibration is quite complex and the willingness to analyze the current state of knowledge regarding the influence of this methodology in the field of neuroplasticity aimed at the recovery of neurological pathologies is appreciable. It would have been interesting to also analyze the endocrinometabolic effects and the aspects of modulation of the muscle quality related to the fmv, which could in turn manifest indirect effects on the neoplasticity induced by the method (although, of course, I realize that this would have made the paper perhaps longer and dispersive). Beyond that, I have not detected any particular critical issues to report regarding the form and content of this paper.

Author Response

Dear reviewer,

We would like to thank you for the effort you put into improving our manuscript. We have found your comments very useful to help us to rework and improve our manuscript. Please, find below our answers and correction to the point raised. We hope that our reply could properly address your concerns and suggestions.

The paper is well presented and seems quite interesting. The topic of focal muscle vibration is quite complex and the willingness to analyze the current state of knowledge regarding the influence of this methodology in the field of neuroplasticity aimed at the recovery of neurological pathologies is appreciable. It would have been interesting to also analyze the endocrinometabolic effects and the aspects of modulation of the muscle quality related to the fmv, which could in turn manifest indirect effects on the neoplasticity induced by the method (although, of course, I realize that this would have made the paper perhaps longer and dispersive). Beyond that, I have not detected any particular critical issues to report regarding the form and content of this paper.

R: We thank the reviewer for these interesting comments. This suggestion about the endocrine-metabolic effects of vibration represents a brilliant hint to deepen the reasoning about the effects of fMV. However, to explore it in full we would extend the review of a large section. We have added a short sentence to mention this aspect to readers and hopefully will cover this subject with the needed space in a further review regarding muscle and peripheral effects of vibration.

We inserted the following sentence at the end of chapter 2.2: “Beyond brain and spinal cord changes, vibration can alter muscle homeostasis affecting directly several endocrine-metabolic parameters. Most of this evidence derives from whole-body vibration paradigms (Di Loreto 2004, Elmantaser 2012), but it could be inferred that to some extent it could occur also after focal muscle vibration.” 

Reviewer 2 Report

Thank you to the authors for this interesting review. It is very well executed. It covers all topics that I would deem needed to an appropriate extend with all needed detail and provides a well-rounded summary of the issue.

I do have quiet a few comments but all of them are related to specific details or argument and should be easy to fix.

Page 2 line 54: This is a miss representation of the reference in my opinion. True recovery is not being contrasted to improvements of daily living. The contrast is compensation. I also do not see the link between compensation and ADL improvement established. The clear connection with neuroplasticity also does not seem to be there. A very weak paragraph all together. This needs fixing.

Page 2 line 82: Some of the information in the following two paragraphs seems beyond the scope and unrelated to your manuscript. I do agree with your next sentence after the two paragraphs. “To describe in detail all the mechanisms underlying brain plasticity goes beyond our purpose”. I would argue even without detail its beyond it.  Reduce to relevant information.

Page 3 line 124 The conclusion of this paragraph seems a little oversimplistic by itself as it disregards the interpersonal variability of stroke. Yes, targeting ipsilesional M1 should usually have the most effect and on average you would not want too much compensatory activity but what about individuals where all of M1 is basically only a lesion? Depending on the stroke there might not be a binary choice between true recovery and compensation. Compensation might be the only choice. This paragraph is not wrong but it seems a little bit too definite on something that is much more complex and unknown.

Page 4 line 152 I find evaluations like here “top-notch”, inappropriate in scientific writing. Provide the attributes of fMV that make it top-notch and leave it to the reader to come to that conclusion. Same for line 160 Interestingly.

Page 4 line 160 expand this part a little bit. How much of a follow up improvement do you have with fMV.

Page 4 line Just present the research and let it speak for itself and let the reader identify it therefor as milestone studies. No need to hype up any of the studies. Especially, in case there is any connection to the authors. I did see that there are common authors in Marconi studies and this research. You do not want there to be any impression you are favoring your own research in an objective review.

Page 4 line 189 I do not understand the drawn connection between afferent input reaching M1 and reorganization. This wording does not make sense. Reorganization is something that would/could happen over time. I can see such a conclusion maybe based on the information in the following sentences but for this sentence it seems wrong.

Page 5 line 207 unclear what the difference between the two adjectives is. Why different and differently specialized. Differently specialized is also different.

Page 5 line 213 I am not sure this is that surprising or poorly understood. The unaffected hemisphere is very important with different functions for motor function and recovery of the affect limb so I do not find this relationship that shocking.

Page 6 line 296 I do not think this conclusion makes sense. Neuroplasticity is also possible well beyond 1 year but that is not what the critical time window is looking for. This is looking for the highest potential for neuroplasticity not any plasticity. So these findings seem to be unrelated to the point you were making earlier.

Page 6 line 324 What point are you making with this study? What is the take away from this finding for fMV before 60 days?

Page 8 line 362 junky wording connecting the two sentence with on the one hand and on the other hand. Reword, it’s hard to read.

Page 8 line 371 missing what you did presented for upper extremity. Some rational for why certain muscle might be the best target for fMV over other muscles.

Page 8 line 391 I am not sure this statement is supported based on you saying there is no agreement. Does improvement scale completely linear endlessly? If I do 12 weeks of fMV compared to 2 weeks will I have 6 times the improvement? There is no plateau? With some plateau other intervention addressing other stroke symptoms might make more sense than endless fMV. More is not always better. It is ok, if we just do not know yet how long fMV should be used.

Page 9 line 521 For a review article there should be some conclusion here. What do you take away from these contrasting approaches. Is this just completely unclear, then provide some specific information that is missing to make the determination if TVR activation is needed or not. Does one of the approaches seem more likely to make sense at the moment and is mostly employed by fMV research? If so why?

Missing Information 1: I am missing some information on the specific target population for fMV. Who can benefit from fMV? Absolutely everyone with a stroke in absolute all stages of stroke? No contraindication? Is there any subpopulation that can especially benefit from stroke?

Missing Information 2: What role do you see for fMV in standard of care? Would fMV be something that would be employed in addition to all standard interventions a stroke survivor receives? Within their standard of care when would be the best moment to employ fMV? Right before a physio or occupational therapy session, right after? Separate from it?  Instead of some standard therapy? Are there studies that compared groups with standard of care and fMV?

All the best!

The Reviewer

See comments to authors, no comments beyond that.

Author Response

Dear reviewer,

We would like to thank you for the effort you have put into improving our manuscript. We have found your comments very useful to help us to rework and improve our manuscript. Please, find below our answers and correction to each point raised. We hope that our reply could properly address your concerns and suggestions.

Thank you to the authors for this interesting review. It is very well executed. It covers all topics that I would deem needed to an appropriate extend with all needed detail and provides a well-rounded summary of the issue.

I do have quiet a few comments but all of them are related to specific details or argument and should be easy to fix.

Page 2 line 54: This is a miss representation of the reference in my opinion. True recovery is not being contrasted to improvements of daily living. The contrast is compensation. I also do not see the link between compensation and ADL improvement established. The clear connection with neuroplasticity also does not seem to be there. A very weak paragraph all together. This needs fixing.

R: Thank you for your comment. Following  your suggestion we rephrased the paragraph as follows

 “Motor recovery after stroke consists of the improvement in two domains. The first is the so-called “true recovery”, which refers to the improvement of body function and structures, and the second is the compensation, which indicates the patient’s ability to accomplish a goal through adaptation of remaining elements or substitution with a new approach. While the latter also depends on implicit and explicit compensatory strategies, true recovery mainly depends on neuroplasticity, which consists of the brain’s natural property to reorganize, changing properties, structures, and pathways to acquire or improve skills.

Page 2 line 82: Some of the information in the following two paragraphs seems beyond the scope and unrelated to your manuscript. I do agree with your next sentence after the two paragraphs. “To describe in detail all the mechanisms underlying brain plasticity goes beyond our purpose”. I would argue even without detail its beyond it.  Reduce to relevant information.

R: Thank you for the suggestion, we deeply reduced the two paragraphs as follows: “After a brain injury such as a stroke, the plasticity-based attempt to recover the function loss involves three distinct but partially overlapping phases: reversal of diaschisis with cell genesis and repair, functional neuronal plasticity (e.g. changing properties of central monoaminergic neuronal pathways), and neuroanatomical plasticity (i.e. the capability to establish and consolidate new neural networks in response to a change in the environment). Even though these changes occur both at a cortical and a peripheral level, they mostly involve the cortex, the ideal site for the plasticity to take place”.

Page 3 line 124 The conclusion of this paragraph seems a little oversimplistic by itself as it disregards the interpersonal variability of stroke. Yes, targeting ipsilesional M1 should usually have the most effect and on average you would not want too much compensatory activity but what about individuals where all of M1 is basically only a lesion? Depending on the stroke there might not be a binary choice between true recovery and compensation. Compensation might be the only choice. This paragraph is not wrong but it seems a little bit too definite on something that is much more complex and unknown.

R: This paragraph originally included the review of all fMRI/TMS/Spect studies aimed at addressing the role of the secondary motor areas (both in the affected and the unaffected hemisphere) in guiding the motor recovery after stroke, but we had to reduce it. As suggested, we rephrased the sentence with: “ipsilesional M1 became one of the most effective targets for rehabilitation therapy”, to “mitigate” the previous statement on the role of ipsilesional M1 modulation that, we agree with you, it looked “too defined” without the cut part. Thank you for your comment indeed.

Page 4 line 152 I find evaluations like here “top-notch”, inappropriate in scientific writing. Provide the attributes of fMV that make it top-notch and leave it to the reader to come to that conclusion. Same for line 160 Interestingly.

R: inappropriate words have been fixed

Page 4 line 160 expand this part a little bit. How much of a follow up improvement do you have with fMV.

R: We want to thank the reviewer for the precious advice. This is a pivotal point even though it is very hard to summarize all the studies on this issue without making the manuscript difficult to be followed. As suggested, we expanded the paragraph as follows:

“Both the clinical improvement and the related neurophysiological changes induced by the fMV treatment outlast the mere period of stimulation itself, being reported to be still present after 30 minutes and several months after the stimulation, respectively, depending on the adopted stimulation protocol (e.g. vibration frequency and amplitude, duration and the number of sessions), the chosen evaluation parameter (e.g. cortical area map/volume, H-reflex and MEP amplitude, etc.) and the outcome evaluated (e.g. improvement in strength, step symmetry, spasticity, discriminative ability, motor planning, execution, etc).”

Page 4 line Just present the research and let it speak for itself and let the reader identify it therefor as milestone studies. No need to hype up any of the studies. Especially, in case there is any connection to the authors. I did see that there are common authors in Marconi studies and this research. You do not want there to be any impression you are favoring your own research in an objective review.

R: Thank you for the suggestion. We fixed this aspect.

Page 4 line 189 I do not understand the drawn connection between afferent input reaching M1 and reorganization. This wording does not make sense. Reorganization is something that would/could happen over time. I can see such a conclusion maybe based on the information in the following sentences but for this sentence it seems wrong.

R: We agree with the reviewer. The sentence was a misprint that referred to subsequent information and has been deleted, we apologize for that.

Page 5 line 207 unclear what the difference between the two adjectives is. Why different and differently specialized. Differently specialized is also different.

R: redundant adjectives fixed.

Page 5 line 213 I am not sure this is that surprising or poorly understood. The unaffected hemisphere is very important with different functions for motor function and recovery of the affect limb so I do not find this relationship that shocking.

R: Thank you for your comment, we fixed the sentence.

Page 6 line 296 I do not think this conclusion makes sense. Neuroplasticity is also possible well beyond 1 year but that is not what the critical time window is looking for. This is looking for the highest potential for neuroplasticity not any plasticity. So these findings seem to be unrelated to the point you were making earlier.

R: We agree with the reviewer. As suggested, we modified the paragraph as follows:

“A great part of the studies aimed at investigating fMV effects on post-stroke motor recovery include patients treated in the late-chronic phase of the disease (≥ 1 year from stroke). Since they widely reported significant fMV-related improvement of several motor aspects, including strength, spasticity, gait, and kinematics parameters, they indirectly show that neuroplasticity is still present 1 year after stroke.”

Page 6 line 324 What point are you making with this study? What is the take away from this finding for fMV before 60 days?

R: Thank you for raising this point that relates to one of the main aspects of the review also further discussed in the last chapter. To date, the vast majority of the published trials are built to investigate the fMV effect on motor recovery by starting the fMV treatment on average 1 year after stroke. Considering that adaptative plastic changes within the motor network that are responsible for motor recovery occur soon after stroke, and that most of the post-stroke rehabilitation guidelines suggest starting the rehabilitation program as soon as possible, in our opinion understanding if fMV is effective even before 60 days is one of the main perspectives of this research field, and the reported randomized controlled trial is, to our knowledge, the sole trial on fMV performed in the acute phase of stroke. This is not surprising, because treating stroke patients in the acute phase presents several difficulties and controversies due to early maladaptive plastic changes that might hinder functional recovery. Anyway, data from the CPASS trial showed smaller but significant improvement in the upper limb recovery of stroke patients treated in the acute phase (i.e. ≤30 days from stroke), and even the Authors concluded that there might be a point in the acute phase where stroke patients could benefit from rehabilitative treatments. Thus, in our opinion, reporting evidence of a motor improvement in stroke patients treated in the acute phase could suggest further RCTs to be designed to individuate a “critical therapeutical point” in the acute phase where it is possible to early act on plastic mechanisms through fMV to maximize post-stroke motor recovery.

Page 8 line 362 junky wording connecting the two sentence with on the one hand and on the other hand. Reword, it’s hard to read.

R: Thank you for the advice, the paragraph has been rephrased (please, see the next comment)

Page 8 line 371 missing what you did presented for upper extremity. Some rational for why certain muscle might be the best target for fMV over other muscles.

R: Thank you for the suggestion. We highlighted the information for the upper extremity that was already present in the first part of the paragraph as follows:  “Regarding the upper limb treatment, several studies applied fMV over the wrist flexors muscles alone or in combination with synergic muscles such as the biceps brachialis, over the biceps brachialis coupled with the pectoralis minor or over the hand. To improve flexor muscle spasticity, other studies applied fMV over extensor muscles, such as the triceps brachii alone or in association with the extensor carpi radialis.”

Page 8 line 391 I am not sure this statement is supported based on you saying there is no agreement. Does improvement scale completely linear endlessly? If I do 12 weeks of fMV compared to 2 weeks will I have 6 times the improvement? There is no plateau? With some plateau other intervention addressing other stroke symptoms might make more sense than endless fMV. More is not always better. It is ok, if we just do not know yet how long fMV should be used.

R: We agree with the reviewer that “more is not always better”. The sentence “the longer the stimulation period the higher the motor control” actually referred to the studies reported in the review, more than to a hypothetical neverending treatment. In this regard, to our knowledge, no comparison studies have been performed to address which is the most effective duration of treatment so far, and data on this issue are very heterogeneous. Anyway, the sentence was quite misleading, thank you for your comment indeed, so we rephrased the paragraph as follows,.

“In this regard, no comparison study has been performed to address which is the most effective duration of treatment. However, since the amount of the induced change grows gradually as conditioning trials continue over subsequent days and weeks, probably the longer the stimulation period, the higher the motor recovery. Moreover, prolonged stimulation can allow the activation and transduction of proteins, and the repetition over consecutive days allows optimal memory consolidation and long persistence of the training-independent sensory learning”.

Page 9 line 521 For a review article there should be some conclusion here. What do you take away from these contrasting approaches. Is this just completely unclear, then provide some specific information that is missing to make the determination if TVR activation is needed or not. Does one of the approaches seem more likely to make sense at the moment and is mostly employed by fMV research? If so why?

R: Thank you for raising this point. This is a strongly debated issue and a definitive conclusion have not been drawn so far, so we preferred not to have a position statement and report both the point of view. Anyway, even though there are no studies aimed at addressing this specific issue, in our opinion co-contraction due to TVR elicitation may negatively impact fMV effects. Moreover, most of the studies adopted a stimulation protocol (i.e. 100 Hz frequency and 0.2-0.5 amplitude) that can avoid the TVR. Thus, following your suggestion, we conclude the paragraph as follows:

“Even though there are no studies aimed at addressing this specific issue, most of the studies adopted a stimulation protocol (i.e. 100 Hz frequency and 0.2-0.5 amplitude) that can avoid the TVR”. 

Missing Information 1: I am missing some information on the specific target population for fMV. Who can benefit from fMV? Absolutely everyone with a stroke in absolute all stages of stroke? No contraindication? Is there any subpopulation that can especially benefit from stroke?

Missing Information 2: What role do you see for fMV in standard of care? Would fMV be something that would be employed in addition to all standard interventions a stroke survivor receives? Within their standard of care when would be the best moment to employ fMV? Right before a physio or occupational therapy session, right after? Separate from it?  Instead of some standard therapy? Are there studies that compared groups with standard of care and fMV?

R: We would like to thank you for these suggestions. We think that adding these missing points greatly improved our manuscript. Except for the stages of stroke that have been already extensively treated, we added the requested information in the last chapter as follows:

“Even though most of the studies have addressed fMV effects in combination with conventional rehabilitation programs, findings in chronic stroke patients were also detected in the absence of any other physical therapy”.

“Patients are usually required to perform at least a minimal isometric voluntary contraction but, in general, fMV has no reported contraindication or side effects, and represents a safe and well-tolerated non-invasive brain stimulation (NIBS) technique, easy to perform at the bedside and able to promote post-stroke motor recovery in every stage of the disease.”

“Few studies have investigated differences in the fMV-related motor improvement depending on stroke characteristics, such as stroke type (i.e. ischemic or hemorrhagic stroke), localization (cortical or subcortical), extension, and side of the lesion, which could influence the efficacy of the fMV treatment. Thus, is still unclear, to date, if there is a stroke subpopulation that can especially benefit from fMV treatment”.

Reviewer 3 Report

The authors did an excellent job. The topic is of particular interest to the scientific community and the review satisfactorily summarises the main arguments. I have no further comments to add. The only suggestion for the authors is to add a section briefly describing the search strategy used. I am well aware that this is a narrative review and the search strategy is not a requirement, however it would make it easier for the reader to find the included articles.

Minor editing of English language required.

Author Response

Dear reviewer,

We would like to thank you for the effort you have put into improving our manuscript. We have found your comments very useful to help us to rework and improve our manuscript. Please, find below our answers and correction to the point raised. We hope that our reply could properly address your concerns and suggestions.

The authors did an excellent job. The topic is of particular interest to the scientific community and the review satisfactorily summarises the main arguments. I have no further comments to add. The only suggestion for the authors is to add a section briefly describing the search strategy used. I am well aware that this is a narrative review and the search strategy is not a requirement, however it would make it easier for the reader to find the included articles.

We thank the reviewer for this suggestion. Indeed, being a narrative review pointing to deepen the mechanism behind the effects of fMV, we did not follow a systematic approach to retrieve articles. Following the suggestion of the reviewer, to help the reader to put data in the correct contest, we specified the time of the information retrieved by the article and the research strategy as follows:

Appendix A:Articles were reviewed without time limits on several databases (PubMed, EMBASE, Scopus, Web of Science, and the Cochrane Library) until March 2023 for fMV. The search strategy included the following terms: (local OR focal OR repetitive) AND (vibration OR muscle vibration) AND (Stroke OR ischemic OR hemorrhagic). Other relevant articles were selected by manual identification within references of screened publications. Unpublished articles were not considered in this study”.